# Dealing with inconclusive SARS-CoV-2 PCR samples—Our experience

Zhivka Stoykova[1,2], Tsvetelina Kostadinova[2,3], Tatina Todorova [1,2]*, Denis Niyazi[1,4], Milena Bozhkova[1,4], Svetomira Bizheva[4], Temenuga Stoeva[1,4]

1 Department of Microbiology and Virology, Medical University Varna, Varna, Bulgaria, 2 Laboratory of Virology, University Hospital "St. Marina", Varna, Bulgaria, 3 Medical College, Medical University Varna, Varna, Bulgaria, 4 Laboratory of Microbiology, University Hospital "St. Marina", Varna, Bulgaria

* Tatina.Todorova@mu-varna.bg

## Abstract

### Purpose

Early confirmation of SARS-CoV-2 is a key point in the timely management of infected patients and contact persons. Routine diagnostics of COVID-19 cases relies on RT-PCR detection of two or three unique sequences of the virus. A serious problem for the laboratories is how to interpret inconclusive samples which are positive for only one of the SARS-CoV-2 specific genes.

### Materials and methods

A total of 16364 naso-oropharyngeal swabs were collected and tested with SARS-CoV-2 Real-TM kit (Sacace Biotechnologies, Italy) between May and September 2020. We retrospectively analyzed their amplification plots to determine the number of inconclusive samples. We also reviewed the medical records to summarize the patient's COVID-19 testing history and basic demographic characteristics.

### Results

We obtained 136 (0.8%) inconclusive samples with amplification signal only for the N-gene. Thirty-nine of the samples were excluded from further analysis as no additional data were available for them. Of the rest of the samples, the majority– 48% (95% CI 38–59%) had a previous history of SARS-CoV-2 positivity, 14% (95% CI 8–23%)–a subsequent history of positivity and 37% (95% CI 28–48%) were considered as false positive.

### Conclusion

A substantial proportion of the inconclusive results should be considered as positive samples at the beginning or the end of the infection. However, the number of false-positive results is also significant and each patient's result should be analyzed separately following the clinical symptoms and epidemiological data.

**Data Availability Statement:** All relevant data are within the paper and its Supporting Information files.

**Funding:** The authors received no specific funding for this work.

**Competing interests:** The authors have declared that no competing interests exist.

## Introduction

Since the end of 2019, we have witnessed an unprecedented public health crisis [1, 2]. In less than six months, the new coronavirus disease (named COVID-19), caused by the severe acute respiratory syndrome coronavirus 2 (SARS-CoV-2) has turned from an exotic and barely known infection to one of the most discussed and analyzed. The disease is still hardly identified because of the diverse clinical picture–symptoms (if present) can mimic other respiratory viral diseases and vary widely from totally asymptomatic cases to life-threatening pneumonia and respiratory failure [3, 4]. This, together with the aggressive transmission of the infection, makes the virological testing of suspicious individuals the most robust tool for precise COVID-19 management.

In the field of laboratory diagnosis, the real-time polymerase chain reaction (RT-PCR) of nasopharyngeal and/or oropharyngeal swabs has been rapidly accepted as the gold standard for accurate and timely control of the infection [5]. Th technique has been considered decisive for both confirmation of symptomatic cases and screening of contact individuals, especially during the first months of the pandemic before the extensive introduction of SARS-CoV-2 antigen and antibody detection tools [6]. Despite the general agreement for the preferred testing method, there are still some controversies about result interpretation of the RT-PCR assays. The majority of commercially available kits rely on the detection of two, three (or even four) SARS-CoV-2 genes to reduce the risk of false-negative results. Preferred for their unique sequences are regions of the E, RdRp, N and ORF1A genes (for a review on available RT-PCR techniques see [7]). The question of how to interpret reactions with an amplification signal of one of the target genes is currently unclear. According to most of the guidelines and operation manuals, when only one specific SARS-CoV-2 gene is detected, especially at the end of the reaction (high cycle threshold (Ct) values) the result is inconclusive and the patient needs retesting [5, 6]. Thus, the test duration increases and the diagnostic decision is delayed. Moreover, the uncertain results pose questions about the sensitivity and specificity of the RT-PCR assay, as any conclusion for the status of the patient could be wrong and introduce additional risk.

The presence of samples positive for only one SARS-CoV-2 specific gene attracted our attention and we decided to analyze how frequent were inconclusive SARS-CoV-2 samples in the practice. We also tried to find the most feasible interpretation of such inconclusive results.

## Materials and methods

This was a retrospective study of the SARS-CoV-2 RT-PCR results obtained between 01.05.2020 and 30.09.2020 in the Virology laboratory of St. Marina University Hospital, Varna, Bulgaria. As one of the first accredited SARS-CoV-2 laboratories in the country, it has been testing samples from most of the hospitals and regional Health Inspectorates in North-East Bulgaria. Between March and December 2020, it was the principal COVID-19 laboratory for the whole territory of North-East Bulgaria serving a total population of around 1351621 (according to the National Statistical Institute of Bulgaria, 2020 estimation). We were routinely testing samples from hospitalized patients and health care workers that showed respiratory infection symptoms, as well as from ambulatory individuals who were contact persons of verified COVID-19 cases. Asymptomatic no-contact persons who needed RT-PCR result for travelling or working purposes were not investigated.

For this study, we analyzed the frequency of inconclusive SARS-CoV-2 samples during the first wave of COVID-19 in Bulgaria (from May 2020 to September 2020) [8]. During this period, the initial criteria of WHO for releasing COVID-19 patients from isolation [9] were followed in the country (see also Order № РД-01-371/30.06.2020 of Bulgarian Ministry of

Health). To discharge people from isolation, the protocol required two negative RT-PCR results on samples taken at least 24 hours apart. As a result, positive individuals accumulated several PCR results and we were able to analyze the "history of testing" and duration of positivity for most of them. With the pandemic expansion, the initial recommendation has become extremely difficult to follow and it was changed (Order № РД-01-604/13.10.2020 of Ministry of Health)–since October 2020 most of the suspicious patients were tested only once.

During the above-mentioned period, a total of 16364 naso-oropharyngeal swabs were collected from hospitalized, out-of-hospital patients or contact individuals and transported to the laboratory in saline water or viral transportation medium.

RNA extraction was performed with SaMag Viral Nucleic Acid Extraction Kit using SaMag-12 instrument (Sacace Biotechnologies, Italy). The initial extraction volume was 400 μL and the elution volume– 50 μL. Isolated RNAs were amplified with SARS-CoV-2 Real-TM kit (Sacace Biotechnologies, Italy). This test allows the reverse transcription and amplification to be performed in a single, one-step reaction and detects three genes: a region of the E gene common for all SARS-like coronaviruses (Fam channel), E and N genes specific for SARS-CoV-2 (Rox and Cy5 channels). To exclude inhibition of RNA extraction and amplification, an internal control RNA is also amplified and detected in the Hex channel. The sample is considered as SARS-CoV-2 positive if there is an amplification signal with defined Ct value for SARS-like coronaviruses and E- and (or) N-genes of SARS-CoV-2; positive for other coronaviruses if amplification is detected only in the Fam channel and inconclusive if there is an amplification signal for only one of the specific SARS-CoV-2 genes. When this was the case, a new sample was requested from the corresponding clinical unit or health inspectorate and the test was repeated in 48 hours. Samples without identified fluorescence signals in all channels, including that for the internal control, were considered invalid and the test was repeated starting from the extraction step. The quality of each RT-PCR assay was confirmed by valid positive and negative controls. Sacace Biotechnologies sets the analytical sensitivity of the assay to 500 copies viral RNA/ml and the diagnostic sensitivity and specificity to 100%.

All data were retrospectively analyzed using the assigned laboratory number for each patient and no additional sampling or patient intervention was performed. Only basic demographic data (age, gender and hospitalization status) were retrieved from the laboratory information system and examined for each patient because of the heterogeneity of the samples– they were obtained from different hospitals and regional health inspectorates in North-East Bulgaria, and clinical data were often incomplete. No sensitive personal data (names, ethnicity, disease outcome, etc.) were used.

Descriptive statistics–Kruskal-Wallis analysis of variance and Mann-Whitney U test for independent measures, and Chi-square test for categorical variables–were the preferred statistical methods with a significance level of 0.05. To test the possible association between the overall positivity rates and the number of inconclusive samples, the Pearson correlation coefficients were calculated. The statistical program used was the R project for statistical computing (version 4.0.4/2021-02-15).

Ethical approval was obtained from the Ethics Committee of the Medical University Varna (Protocol Approval Number 114).

## Results

For the period between May 2020 and September 2020, we tested 16364 samples collected from COVID-19 suspicious individuals or contact persons. The proportion of positive results with N and E SARS CoV-2 specific genes amplified was 12.6% (2054 out of 16364). A total of 136 naso-oropharyngeal swabs (0.8%) had shown a positive amplification signal only for the N

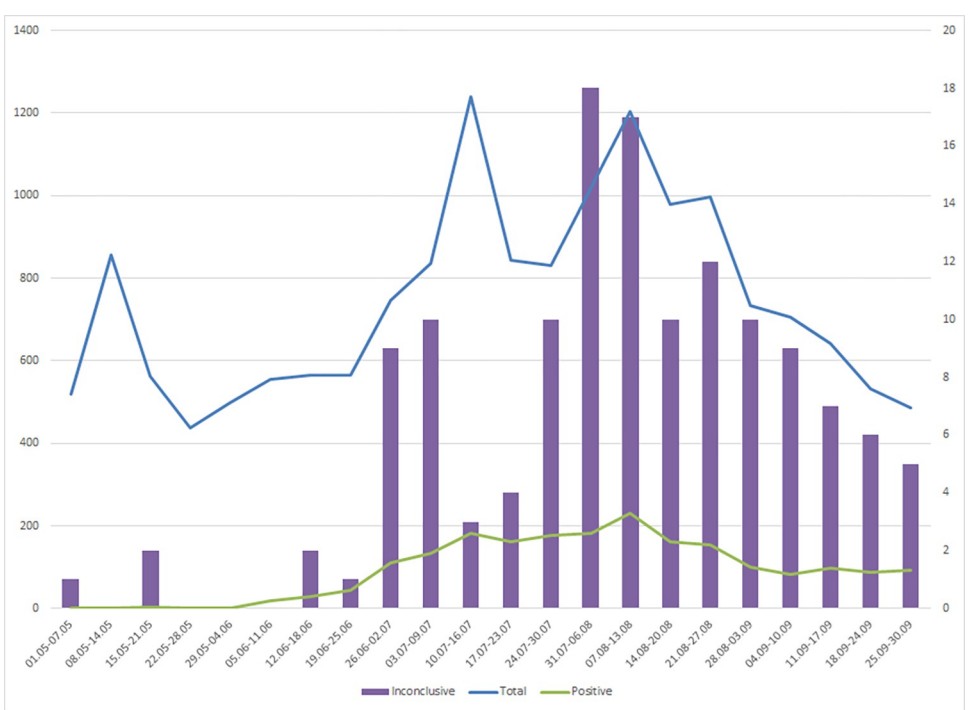

**Fig 1. Weekly distribution of SARS-CoV-2 positive and inconclusive samples between May and September 2020.**
The blue line represents the total number of tested samples (left vertical axis); the green line–the positive cases (left vertical axis), and the purple bars–the number of inconclusive SARS-CoV-2 samples (right vertical axis).

gene with an average Ct of 32.8 (range 27.7–38.2). We did not detect samples with isolated amplification of the E gene.

Fig 1 shows the dynamics of the PCR testing and the change in the positive rate during the studied period. The number of inconclusive samples correlated well with both the total number of tested samples (Pearson correlation coefficient of 0.64, p = 0.001) and the rate of SARS-CoV-2 positivity (Pearson correlation coefficient of 0.80, p < 0.001).

A short description of the demographic characteristics of the groups of patients with the positive, negative and inconclusive results is shown in Table 1. The sex ratio, the proportion of hospitalized patients and the age distribution were similar between the groups of patients with positive and inconclusive samples: 56% females (95% CI 54 to 58%) vs. 57% females (95% CI 49 to 66%); 65% hospitalized patients (95% CI 63 to 67%) vs. 61% (95% CI 52 to 69%) and mean age of 53 years (95% CI 52 to 54) vs. 54 years (95% CI 51 to 57), respectively.

**Table 1. Basic demographic characteristics of individuals with positive, negative and inconclusive SARS-CoV-2 RT-PCR results (01.05–30.09.2020).**

|  | Positive samples (N = 2054) | Negative samples (N = 14174) | Inconclusive samples (N = 136) |
|---|---|---|---|
| **Sex** | | | |
| Female | 1146 (56%) | 9127 (64%) | 78 (57%) |
| Male | 908 (44%) | 5047 (36%) | 58 (43%) |
| **Age** | | | |
| Mean±SD | 53±20 | 51±19 | 54±20 |
| Median | 56 | 53 | 55 |
| **Hospitalization** | | | |
| Yes | 1332 (65%) | 7750 (55%) | 83 (61%) |
| No | 722 (35%) | 6424 (45%) | 53 (39%) |

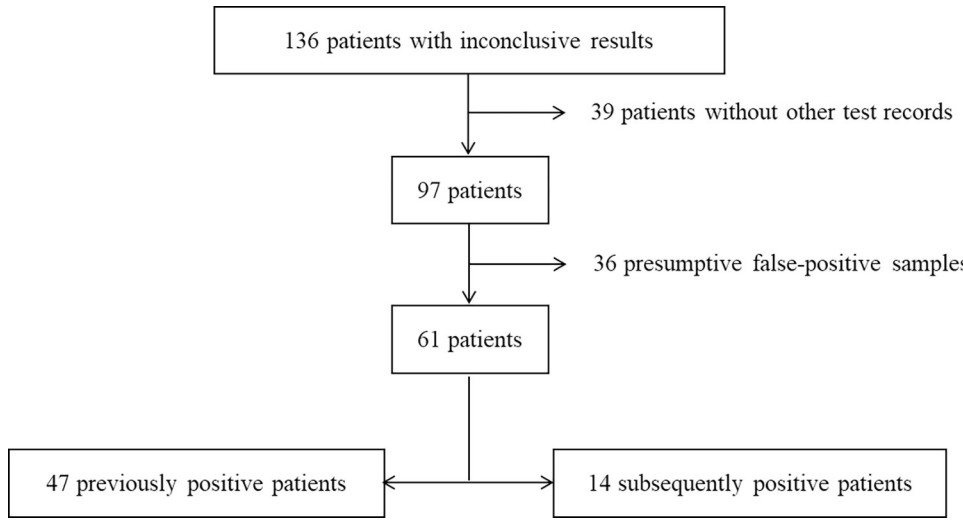

**Fig 2. Patients' flowchart.**

The analysis of the inconclusive samples (Fig 2) showed that for 39 patients, the RT-PCR test with the inconclusive result was the only one recorded in the database and we excluded them from further analysis. For the rest 97 patients, at least one additional sample was obtained and analyzed after the inconclusive result. For 36 of these patients, no history of positive tests or hospital admission existed and the testing of the new sample did not show any amplification–we considered them as samples with false-positive signal for the N gene. Among the other 61 patients, 47 had one or several previous positive results, while 14 –one or several subsequent positive results. The inconclusive result was the very last or first in their SARS-CoV-2 positive testing history. Only two samples were associated with patients with both previous and subsequent positive tests. As the previous positivity was significantly longer than the positivity after the inconclusive result (13 versus 7 and 8 versus four days, respectively, see also S2 Table), we considered them into the group of patients with previous positive results.

The interval between the first positive sample and the inconclusive result varied widely–from 6 to 38 days with an average of 14 days. A large timeframe was also detected for the individuals who had an inconclusive result and then one or more positive results–subsequent positivity lagged from 1 to 29 days with an average of 11 days. The Mann-Whitney U test revealed that the difference between a randomly selected value of the group with the previous positivity history and the group with subsequent positivity was not big enough to be statistically significant (p = 0.17).

The average Ct detected among the samples with previous SARS-CoV-2 positivity was 32.7 ±1.8; among the samples with subsequent positivity– 32.9±1.5 and among the samples with presumably false positive signal– 32.9±1.7 (p = 0.63).

## Discussion

From May to September 2020 (which roughly covers the first wave of COVID-19 in Bulgaria) [8] we detected around 1% of inconclusive RT-PCR tests with amplification of only one of the two SARS CoV-2 specific genes. The proportion of single gene-positive samples differs widely across the scarce literature data with a prevalence rate between 0.3% [10] to 21% [11]. In a study from the Republic of Korea, conducted also in the first months of the pandemic, Lim et al. obtained similar to our results number of inconclusive samples of around 1% [12].

However, comparing the results among the studies is challenging because a wide variety of test systems is currently in use and laboratories are working with different patients groups.

We also tried to clarify the clinical importance of samples positive for one SARS CoV-2 specific gene–if such amplification indicated very high cycle thresholds or was a false-positive result. Our experience tips the balance towards the former interpretation– 48% of the inconclusive samples for the period of the study were obtained from patients who previously had shown positivity for SARS-CoV-2 and 14%—from patients with subsequent positivity. Viral loads are lower before the onset of the symptoms and at the end of the infection [13, 14]. Therefore, the detection of only one of the SARS-CoV-2 specific genes could result from the natural decrease in the amount of SARS-CoV-2 RNA in the course of infection kinetics.

An additional interesting point that merits discussion was that all of the inconclusive results we had obtained were with amplification of the N-gene. This is not surprising as the N protein and the corresponding mRNA are the most abundant in the replication cycle of coronaviruses [15, 16]. Prolonged SARS-CoV-2 positivity has been widely reported in the literature [17–19], as well as positivity before the onset of the clinical symptoms [20]. We presume to speculate that the N-gene was the first and the last to be expressed in the infected cells. The remnant N gene subgenomic RNA, which are abundant in inconclusive samples [12, 21], could be partially amplified in RT-PCR assays causing weak positive signals. The different analytical sensitivity of the RT-PCR assay for the individual genes could also play a role–each primer-probe set has a specific target and even single-point mutations could lead to amplification failure of the corresponding gene [22, 23]. Most likely, a large proportion of the inconclusive results should be considered as positive samples with a high cycle threshold or E gene target failure.

Nevertheless, around 37% of the inconclusive samples were classified as false-positive. Causes for the false positivity could be found in the non-specific binding during the late phases of the PCR reaction [24]–an assumption that is supported by the relatively high Ct values obtained. An additional source for inconclusive results could be contamination of the samples from the other highly positive samples in the same assay. A strong correlation between the rates of inconclusive and positive results existed, and we should admit that during the first months of the pandemic, the burden for rapid and accurate testing was enormous. However, both inconclusive and positive detection rates depended on the total number of tested samples. If the contamination is a real reason for the majority of the uncertain results, someone should expect to have inconclusive samples with the E-gene signal too, which wasn't the case. Inconclusive samples were also detected in weeks when positive samples were not found (Fig 1 and S1 Table).

Finally, pre-analytical factors could also lead to an inconclusive result in an infected person: poor quality of the collected sample (too little material or inappropriate sampling); poor handling and shipment of the specimen; or technical issues, such as inhibition during the assay are well-recognized reasons for an unclear testing conclusion. Pre-analytical causes could explain why inconclusive results were obtained in two cases with previous and subsequent positivity.

To the best of our knowledge, the current work is the sole analysis of this kind in Eastern Europe and one of the rare studies trying to resolve an actual practical issue that many COVID-19 diagnostic laboratories face. However, the study has some limitations, among which the major are its retrospective nature and the short-term period of analysis. We evaluated just the first COVID-19 wave when multiple results were available for most of the positive individuals. In this way, we could not validate our data for all of the SARS-CoV-2 waves and variants detected later. In addition, the number of inconclusive samples stood for less than 1% of all tested samples, and this made the statistical comparison between the groups uncertain. A significant number of the inconclusive samples were also excluded from further analysis

because no additional information was available. This definitely could impact the final results and indicates possible gaps in the management of COVID-19 cases.

## Conclusion

The present study enlightens one of the major problems in the current laboratory identification of SARS-CoV-2 –the inconclusive RT-PCR results received in a time of significant pressure towards the laboratories for rapid and correct diagnostic. In most of the clinical units, algorithms on how to interpret such samples are still missing. Based on our experience, we recommend when it is not possible to repeat the test, to consider inconclusive samples as positive, especially when clinical symptoms are present.

## Supporting information

**S1 Table. List of samples tested for SARS-CoV-2 RNA between 01.05.2020 and 30.09.2020 in the Virology laboratory of St. Marina University Hospital, Varna, Bulgaria.** To ensure fully anonymization the laboratory number of each sample was replaced with an arbitrary given number. Sheet 1 contains the raw data and Sheet2 the weekly summary of the same data. (XLSX)

**S2 Table. List of inconclusive samples obtained between 01.05.2020 and 30.09.2020 in the Virology laboratory of St. Marina University Hospital, Varna, Bulgaria.** To ensure fully anonymization, the laboratory number of each sample was replaced with an arbitrary given number.
(XLSX)

## Author Contributions

**Conceptualization:** Zhivka Stoykova, Tsvetelina Kostadinova, Tatina Todorova, Svetomira Bizheva, Temenuga Stoeva.

**Data curation:** Tatina Todorova.

**Investigation:** Zhivka Stoykova, Tsvetelina Kostadinova, Denis Niyazi, Milena Bozhkova, Svetomira Bizheva.

**Methodology:** Tatina Todorova, Milena Bozhkova.

**Resources:** Temenuga Stoeva.

**Supervision:** Temenuga Stoeva.

**Writing – original draft:** Tsvetelina Kostadinova, Tatina Todorova.

**Writing – review & editing:** Tatina Todorova, Denis Niyazi, Temenuga Stoeva.

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
