## [Decision Letter · Decision Letter 0]

4 Feb 2022

PONE-D-21-31632Dealing with inconclusive SARS-CoV-2 PCR samples – our experiencePLOS ONE

Dear Dr. Тодорова,

Thank you for submitting your manuscript to PLOS ONE. After careful consideration, we feel that it has merit but does not fully meet PLOS ONE’s publication criteria as it currently stands. Therefore, we invite you to submit a revised version of the manuscript that addresses the points raised during the review process. Please pay carefull attention to the reviewers's comment both regarding contextualising the study and accuracy of the terminologies used. In particular, in both the methods and discussion, highlight the population being tested and provide an account of the change in positivity rate of PCR during the study period. Indicate whether positivity rate affect the number and likely cause of inconclusive result and suggest potential solution/interpretation to this changing trends. 

We look forward to receiving your revised manuscript.

Kind regards,

Nei-yuan Hsiao

Academic Editor

PLOS ONE

Journal Requirements:

Reviewers' comments:

Reviewer's Responses to Questions

**Comments to the Author**

1. Is the manuscript technically sound, and do the data support the conclusions?

Reviewer #1: Yes

Reviewer #2: Partly

2. Has the statistical analysis been performed appropriately and rigorously? 

Reviewer #1: Yes

Reviewer #2: I Don't Know

3. Have the authors made all data underlying the findings in their manuscript fully available?

Reviewer #1: Yes

Reviewer #2: Yes

4. Is the manuscript presented in an intelligible fashion and written in standard English?

Reviewer #1: Yes

Reviewer #2: Yes

5. Review Comments to the Author

Reviewer #1: In a retrospective laboratory-based study Stoykova et al. describe inconclusive results from the St. Marina University Hospital, Varna, Bulgaria. 136 of 16364 samples tested inconclusive. These results were further investigated against other tests from the same patients, of these 39 had no additional tests, 47 had prior positive tests, 14 subsequent positive tests and 39 subsequent negative tests.

The authors conclude that most inconclusive results are true positive and should be considered as positive, especially when symptoms are present. The manuscript is of importance to the field and is generally well-written and easy to follow. Tables and figures are appropriate.

I have some comments for improvement:

1. Please provide more detail about the laboratory: Is this a public health or private health laboratory? What population does it serve?

2. The difference in patient characteristics were described as” between the groups of positive and negative samples.” It should however be noted that far fewer inconclusive cases were included which would mean that the analysis might be underpowered when comparing inconclusive with other categories. Line 105 and 106 appear to be a table legend – please indicate this clearly.

3. Please elaborate on the causes of single gene positive inconclusive results including pre-analytical and analytical causes – such as true low viral loads, inhibition, sample contamination, primer mismatches etc. and in the discussion of how to interpret inconclusive results. Please also consider pre-analytic factors that might change over the course of the epidemic such whether patients are seen during an epidemic wave upswing or thereafter, when many more patients may have residual low viral loads during the recovery phase.

4. Please provide data about the time-difference between the initial inconclusive and subsequent tests (for cases that tested positive vs negative on subsequent samples) and in the limitations please consider to discuss that samples that initially tested inconclusive and subsequently negative might have become negative due to the natural kinetics of SARS-CoV-2 RNA in infected patients, with true low viral loads declining to becoming undetectable over time.

Language:

Line 19: Routine diagnostic – replace with “Routine diagnostics”

Line 131: “scare” meaning is unclear; please replace with the correct term, perhaps “scarce”

Reviewer #2: 1. This is an important topic that requires study. There are various different real time PCR assays for SARS-CoV-2 and it has been challenging interpreting and dealing with inconclusive results.

2. Abstract:

Line 20: “detection of at least two unique sequences of the virus” is not entirely accurate, rather use two SARS-CoV-2 specific gene targets or three gene targets.

In results, lines 31-33, the absolute values are stated, it would be preferable to use the percentage with 95% confidence intervals.

It is also noted that a large number of inconclusive results were excluded from the analysis due to lack of data, this could have impacted on results.

3. Keywords line 41-42: viral load is not suitable as a keyword as SARS-CoV-2 PCR is primarily a qualitative test and not reported as a viral load.

4. Introduction: Noted to be quite a general introduction with few references.

Lines 54-57 refers to guidelines and operation manuals but this is not referenced.

Line 58 – Provide clarification as to how detection of single gene targets poses questions on the sensitivity of the RT-PCR assay. Specificity can also be affected as some of these can be false positive.

5. Methods:

Line 65 – why was the timeframe May-September 2020 chosen? Was this during a peak period of COVID-19 or between the waves as this can also impact on testing and subsequent results eg. many samples with high viral loads during the peak can cause cross contamination and false positive single N gene target detection. It is only subsequently noted in the discussion that this period roughly covers the first wave of COVID-19 in Bulgaria (lines 128-129). The PCR positivity rate during this period was noted to be 12.6% (line 92). What was the case definition for testing during this period? – did it also include testing of asymptomatic exposed cases, screening of all hospitalized patients regardless of exposure or symptoms.

Line 86-87 – I am not sure if https://www.socscistatistics.com/ is suitable or acceptable to use for statistical analysis in biomedical studies, however the website indicates that it has been audited for accuracy against the output produced by a number of established statistics packages, including SPSS. Most published studies use established statistical softwares such as STATA, SPSS, etc.

Lines 92-95 – describes the result distribution of the PCR assay. It is also interesting that there were no invalid results with internal control failure. In my experience, beside the issue of inconclusive results, the problem with invalid results has been challenging.

Line 27-28 & 84-85 - Was the demographic data taken from electronic medical records or patient files or from a laboratory information system as there is minimal clinical data.

No ethics application or clearance is noted - Even though the study was a retrospective analysis of laboratory data, ethics approval is still required from the relevant institution as patient data is being used and for publication purposes.

6. Results:

Lines 112-113 – the interval/timeframe between the inconclusive and positive test results would also be useful to be indicated to give an understanding how long and when in the course of infection single N gene positivity can occur.

Line 148 – “Causes for these false-positivity could be found in the different analytical sensitivity of the RT-PCR assay for the individual genes” – please clarify how this can cause false positivity. The analytical sensitivity/limit of detection of the RT-PCR assay used is not mentioned.

Line 150 – what were the Ct values obtained for 37% of inconclusive results that were false positive in comparison to the true positive results (48% of the inconclusive samples from patients who previously had shown positivity for SARS-CoV-2 and 14% from patients with subsequent positivity) and was there a significant difference? A graphical comparison would be useful.

7. Discussion: Noted to be brief.

There is no mention of other factors that can cause inconclusive results such as pre-analytical factors eg. specimen type, quality, handling & storage conditions.

It would be helpful with the interpretation of results to know the background history such as if patient symptomatic or not, onset of symptoms in relation to testing, contact of confirmed case, etc. Several published studies on this topic have reported both clinical and epidemiologic data. This is lacking in this study.

The limitations of the study are not mentioned which is required.

6. PLOS authors have the option to publish the peer review history of their article (what does this mean?). If published, this will include your full peer review and any attached files.

Reviewer #1: **Yes: **Gert U. van Zyl

Reviewer #2: **Yes: **Aabida Khan

---

## [Author Response · Author response to Decision Letter 0]

21 Mar 2022

Dear Editor,

Dear Reviewers,

We are grateful for the positive feedback and the possibility to revise our work. The academic editor and reviewers raised many interesting points and we hope that any concerns have been suitably addressed in the revised version and this letter. The responses to the specific commentaries and criticisms are as follows:

Editor Comments:

Please pay careful attention to the reviewers's comment both regarding contextualising the study and accuracy of the terminologies used. In particular, in both the methods and discussion, highlight the population being tested and provide an account of the change in positivity rate of PCR during the study period. Indicate whether positivity rate affect the number and likely cause of inconclusive result and suggest potential solution/interpretation to this changing trends.

We agree with the editor that all these features are crucial for the improvement of the manuscript – accordingly, we have strengthened different parts of the work to reflect these concerns. The population of interest is now described in the Materials and methods, and the weekly change in the rates of testing and positivity are present in the Results (Fig 1). 

Journal Requirements:

The style requirements were carefully checked and observed in the new version of the manuscript.

2. In your Data Availability statement, you have not specified where the minimal data set underlying the results described in your manuscript can be found. PLOS defines a study's minimal data set as the underlying data used to reach the conclusions drawn in the manuscript and any additional data required to replicate the reported study findings in their entirety. All PLOS journals require that the minimal data set be made fully available. 

The study’s minimal underlying data set is uploaded as Supporting Information files. The Supporting files consist of two tables: S1_Table, which contains the raw data of all samples tested in the studied period and S2_Table, which contains the data associated with the inconclusive RT-PCR samples. All results were retrospectively analyzed using the assigned laboratory number for each patient and no sensitive personal data were used. To ensure fully anonymization, the laboratory number of each sample was replaced with an arbitrary given number before the upload of the data.

We included this phrase in the previous version of the manuscript because the information was not considered to be crucial for the main conclusions of the work. Now, the phrase was removed from the revised version of the paper and the average Ct of the different groups of samples are given in the last paragraph of the Result Section.

The reference list is complete and follows the style of the Journal. Because of the necessary enlargement of Introduction and Discussion Sections (as suggested by the reviewers), 11 new references were included in this Section.

Reviewer 1: 

We thank the reviewer for the generally positive reaction to the paper.

1. Please provide more detail about the laboratory: Is this a public health or private health laboratory? What population does it serve?

We have provided more information about the laboratory. The first paragraph of the Materials and methods Section is now improved with details about the region and the population covered by the laboratory.

2. The difference in patient characteristics were described as” between the groups of positive and negative samples.” It should however be noted that far fewer inconclusive cases were included which would mean that the analysis might be underpowered when comparing inconclusive with other categories.

We realize that the number of the inconclusive samples is significantly lower than the negative and positive results. However, this is the only way to compare their characteristics in a statistically sound manner. In the Discussion of the revised version, this was indicated as a limitation of the study.

 Line 105 and 106 appear to be a table legend – please indicate this clearly.

The Table legend is now separated from the main body of the text.

3. Please elaborate on the causes of single gene positive inconclusive results including pre-analytical and analytical causes – such as true low viral loads, inhibition, sample contamination, primer mismatches etc. and in the discussion of how to interpret inconclusive results. Please also consider pre-analytic factors that might change over the course of the epidemic such whether patients are seen during an epidemic wave upswing or thereafter, when many more patients may have residual low viral loads during the recovery phase.

To address this concern, causes for inconclusive results are now discussed in the Discussion. This section of the manuscript was entirely rewritten and amended with emphasis on the possible reasons for single gene amplification.

4. Please provide data about the time-difference between the initial inconclusive and subsequent tests (for cases that tested positive vs negative on subsequent samples) and in the limitations please consider to discuss that samples that initially tested inconclusive and subsequently negative might have become negative due to the natural kinetics of SARS-CoV-2 RNA in infected patients, with true low viral loads declining to becoming undetectable over time.

This suggestion is of particular importance and we included in the new version the time from the first positive result to the inconclusive one and between the initial inconclusive and the last positive sample. The natural kinetics of SARS-CoV-2 infection is also discussed.

Line 19: Routine diagnostic – replace with “Routine diagnostics”

Line 131: “scare” meaning is unclear; please replace with the correct term, perhaps “scarce”

The language is now corrected.

Reviewer #2: 

1. This is an important topic that requires study. There are various different real time PCR assays for SARS-CoV-2 and it has been challenging interpreting and dealing with inconclusive results. 

We acknowledge the positive feedback and the detailed comments of the reviewer.

2. Abstract:

Line 20: “detection of at least two unique sequences of the virus” is not entirely accurate, rather use two SARS-CoV-2 specific gene targets or three gene targets.

In results, lines 31-33, the absolute values are stated, it would be preferable to use the percentage with 95% confidence intervals.

We agree with the reviewer and the Abstract was changed accordingly.

It is also noted that a large number of inconclusive results were excluded from the analysis due to lack of data, this could have impacted on results.

The concern about the significant number of excluded results is discussed as one of the limitations of the study.

3. Keywords line 41-42: viral load is not suitable as a keyword as SARS-CoV-2 PCR is primarily a qualitative test and not reported as a viral load.

We admit that viral load is not the best choice for a keyword and we had dropped it from the list of the keywords.

4. Introduction: Noted to be quite a general introduction with few references.

We realize that the previous version suffered from a short Introduction and Discussion Section. In the new version, new paragraphs with additional background information were added to make the text more ‘reader-friendly’ and complete.

Lines 54-57 refers to guidelines and operation manuals but this is not referenced.

References are now added.

Line 58 – Provide clarification as to how detection of single gene targets poses questions on the sensitivity of the RT-PCR assay. Specificity can also be affected as some of these can be false positive.

The sentence ‘Moreover, this poses questions about the sensitivity of the RT-PCR assay.’ is now changed to “Moreover, the uncertain results pose questions about the sensitivity and specificity of the RT-PCR assay, as any conclusion for the status of the patient could be wrong and introduce additional risk.”

5. Methods:

Line 65 – why was the timeframe May-September 2020 chosen? Was this during a peak period of COVID-19 or between the waves as this can also impact on testing and subsequent results eg. many samples with high viral loads during the peak can cause cross contamination and false positive single N gene target detection. It is only subsequently noted in the discussion that this period roughly covers the first wave of COVID-19 in Bulgaria (lines 128-129). The PCR positivity rate during this period was noted to be 12.6% (line 92). What was the case definition for testing during this period? – did it also include testing of asymptomatic exposed cases, screening of all hospitalized patients regardless of exposure or symptoms.

The second paragraph of the Materials and methods section now contains more information on why this period was chosen for analysis. We also clarified the groups of tested patients and the region served by the laboratory. 

Line 86-87 – I am not sure if https://www.socscistatistics.com/ is suitable or acceptable to use for statistical analysis in biomedical studies, however the website indicates that it has been audited for accuracy against the output produced by a number of established statistics packages, including SPSS. Most published studies use established statistical softwares such as STATA, SPSS, etc.

Although the listed online calculator is a really good (and free) statistical tool, and the performed statistical tests are quite simple to generate serious inconsistency, we agree that a more rigorous program should be used in science. The same analyses were redone with the R project for statistical computing (version 4.0.4/2021-02-15) software. This led to negligible differences in the obtained results.

Lines 92-95 – describes the result distribution of the PCR assay. It is also interesting that there were no invalid results with internal control failure. In my experience, beside the issue of inconclusive results, the problem with invalid results has been challenging.

We admit that more clarification is needed about our choice to discuss only valid RT-PCR results. The issue of invalid samples is significant for the laboratory practice, but in our experience, it is more related to technical problems during the assay – we have noticed that invalid results are not equally distributed in the time and correlate with the kit lot used for either extraction or amplification. Additionally, the Bulgarian National COVID-19 information system does not allow invalid results to be uploaded (only three options are available – positive, negative or inconclusive result). This leads to completely different management of invalid samples and if such a result is obtained the sample is repeatedly processed starting from the extraction step. Only in case, when the second or the third extraction and amplification are unsuccessful, a new sample is requested from the corresponding clinical or out-of-hospital unit. The new sample has been processed and registered under the laboratory number of the original sample, which means that it is almost impossible to perform correct retrospective analysis for the number of invalid results.

Line 27-28 & 84-85 - Was the demographic data taken from electronic medical records or patient files or from a laboratory information system as there is minimal clinical data.

The demographic data were retrieved from the laboratory information system. In the new Materials and methods section the reasons for the minimal clinical data are now explained – during these first months of the pandemic we have been receiving samples from approximately 20 different hospitals and health inspectorates often with incomplete patient data. 

No ethics application or clearance is noted - Even though the study was a retrospective analysis of laboratory data, ethics approval is still required from the relevant institution as patient data is being used and for publication purposes.

As the study is fully retrospective and all data were retrieved from standard-of-care tests without additional sampling, it was not considered as ethically controversial. However, we admit that ethic approval is necessary for publication of the results and accordingly the manuscript has been submitted to the institutional ethic committee and approved for publication (Protocol Approval Number 114 by the Ethics Committee of the Medical University Varna). The last paragraph of the Materials and methods presents the ethical approval.

6. Results:

Lines 112-113 – the interval/timeframe between the inconclusive and positive test results would also be useful to be indicated to give an understanding how long and when in the course of infection single N gene positivity can occur.

As suggested, a new paragraph discussing the time between the first positive or the last positive result and the inconclusive result is added to the Result section.

Line 148 – “Causes for these false-positivity could be found in the different analytical sensitivity of the RT-PCR assay for the individual genes” – please clarify how this can cause false positivity. The analytical sensitivity/limit of detection of the RT-PCR assay used is not mentioned.

Causes for the false positivity are now more elaborated in the Discussion section. The analytical sensitivity of the RT-PCR kit used is included at the end of the Materials and methods section.

Line 150 – what were the Ct values obtained for 37% of inconclusive results that were false positive in comparison to the true positive results (48% of the inconclusive samples from patients who previously had shown positivity for SARS-CoV-2 and 14% from patients with subsequent positivity) and was there a significant difference? A graphical comparison would be useful.

The Ct values were also added to the main text for the three groups of interest. However, because there was no any significant difference we consider graphical presentation unnecessary.

7. Discussion: Noted to be brief.

There is no mention of other factors that can cause inconclusive results such as pre-analytical factors eg. specimen type, quality, handling & storage conditions.

The Discussion has been completely rewritten and all these causes suggested by the Reviewer are now part of it.

It would be helpful with the interpretation of results to know the background history such as if patient symptomatic or not, onset of symptoms in relation to testing, contact of confirmed case, etc. Several published studies on this topic have reported both clinical and epidemiologic data. This is lacking in this study.

We agree that adding the background history of the patients would improve the quality of our work. Unfortunately, clinical units and regional health inspectorates in Bulgaria have not been obliged to submit the patient’s clinical background to the laboratory which has been tested the corresponding sample. We were unable to collect the complete clinical information and we decided to focus only on the basic demographic characteristics. Please, refer also to the response to the comment about the minimal demographic data of the patients.

The limitations of the study are not mentioned which is required.

We admit that in the previous version of the manuscript we have not focused in detail on the limitations of the study. To address this concern, they are now included at the end of the manuscript.

Once again thank you for your comments and suggestions, which we consider as a significant improvement of our work.

Sincerely yours,

Tatina Todorova

---

## [Editor Report · Decision Letter 1]

25 Apr 2022

Dealing with inconclusive SARS-CoV-2 PCR samples – our experience

PONE-D-21-31632R1

Dear Dr. Тодорова,

We’re pleased to inform you that your manuscript has been judged scientifically suitable for publication and will be formally accepted for publication once it meets all outstanding technical requirements.

Kind regards,

Nei-yuan Hsiao

Academic Editor

PLOS ONE

Additional Editor Comments (optional):

Thank you for amending the manuscript which is now accepted for publication. I suggest minor amendments of:

1) remove the p value column from table 1. If you wish to discuss the difference in population (in text) it is better to use confidence intervals and compare mainly positive vs inconclusive.

2) removing (low) "viral load" and replace them with (high) cycle threshold as assay used is not quantitative. Although Ct is a proxy to viral load there are situations where the two may differ and thus it is better to use a more precise language.
---

## [Editor Report · Acceptance letter]

5 May 2022

PONE-D-21-31632R1 

Dealing with inconclusive SARS-CoV-2 PCR samples – our experience 

Dear Dr. Todorova:

I'm pleased to inform you that your manuscript has been deemed suitable for publication in PLOS ONE. Congratulations! Your manuscript is now with our production department. 

Kind regards, 

on behalf of

Dr. Nei-yuan Hsiao 

Academic Editor

PLOS ONE